# PHYSICS OF LEARNING: A LAGRANGIAN PERSPECTIVE TO DIFFERENT LEARNING PARADIGMS

## ABSTRACT

We study the problem of building an efficient learning system. Efficient learning processes information in the least time, i.e., building a system that reaches a desired error threshold with the least number of observations. Building upon least action principles from physics, we derive classic learning algorithms, Bellman's optimality equation in reinforcement learning, and the Adam optimizer in generative models from first principles, i.e., the Learning *Lagrangian*. We postulate that learning searches for stationary paths in the Lagrangian, and learning algorithms are derivable by seeking the stationary trajectories.

Table 1: Overview of Physics-Inspired Learning *Lagrangian*. Machine learning encompasses a broad set of paradigms from supervised, unsupervised learning to reinforcement learning and generative models. We postulate that learning also follows a physical law, the principle of least action. We unify different learning paradigms through derivation from the first principles. In particular, we compare the learning *Lagrangian* with existing physical laws and detail each principle's suitable application in learning tasks. We derive classical learning algorithms that arise when searching for stationary solutions in the Lagrangian.

| | Physics | Learning |
|---|---|---|
| Fermat's principle | $T = \int_A^B dt$ | $T = \int_{\epsilon[\emptyset]}^{\epsilon[\mathbf{s}]} dt$ [*] |
| *Hamiltonian* | $H(\mathbf{x}, \mathbf{p}) = \mathbf{p} \cdot \dot{\mathbf{x}} - L(\mathbf{x}, \dot{\mathbf{x}})$ | $H(\mathbf{s}, \mathbf{a}, \lambda) = r(\mathbf{s}, \mathbf{a}) + f(\mathbf{s}, \mathbf{a})^T \lambda$ [†] |
| the *Lagrangian* | $L = T - V$ | $L(\ell, \nabla_\theta \ell) = \frac{1}{2}(\nabla_\theta \ell)^T F^{-1} \nabla_\theta \ell - \ell(\theta)$ [*] |

| | Applications | Algorithms |
|---|---|---|
| Fermat's principle | Parametric Models | A-optimality (Atkinson et al., 2007) |
| *Hamiltonian* | Reinforcement Learning | Bellman's Equation (Bellman, 1958) |
| the *Lagrangian* | Generative Models / Supervised Learning | Adam (Kingma, 2014) / RMSprop (Tieleman, 2012) |

*Notes:* $T$ in Fermat's principle denotes time taken to travel from point $A$ to point $B$; $\epsilon[\emptyset], \epsilon[\mathbf{s}]$ is the generalization error after observing zero data to data sequence $\mathbf{s} := s_1, s_2, \ldots$; $H$ is the (physical) Hamiltonian system with position $\mathbf{x}$ and momentum $\mathbf{p}$ and Lagrangian $L$; $H(\mathbf{s}, \mathbf{a}, \lambda)$ is the reinforcement learning correspondent with state $\mathbf{s}$, action $\mathbf{a}$, reward $r(\mathbf{s}, \mathbf{a})$, transition dynamics $f(\mathbf{s}, \mathbf{a})$ and momentum equivalent $\lambda$; $L = T - V$ represents kinetic energy minus potential energy; $\ell$ denotes some log-likelihood function; $\nabla_\theta \ell$ is gradient with respect to model parameters $\theta \in \mathbb{R}^P$; $F^{-1}$ denotes the inverse Fisher information. Bold symbols are vectors; $(\cdot)^\top$ is transpose; $\dot{x}$ is derivative with respect to time. The learning Lagrangian indicated via [†] means it is classic textbook material in control theory (see Todorov (2006)). Learning Lagrangians indicated by [*] are proposed in this work; to the best of our knowledge, no prior published work exists as of September 2025.

## 1 INTRODUCTION

Modern machine learning encompasses a broad set of paradigms — supervised and unsupervised learning, reinforcement learning, and generative models, with deep architectures as the dominant modeling substrate. As momentum built across labs, industry, and policymakers, work shifted toward translating technical advances into products. These efforts have accelerated deployment but also privileged trial-and-error engineering and scale-first heuristics, in part because we still lack a principled understanding of *when and why* learning emerges, generalizes, and fails. This gap has impeded a systematic methodology for designing sample- and compute-efficient learning systems.

This paper demonstrates a close connection between physics and learning and postulates that learning algorithms arise as stationary trajectories of a learning *Lagrangian*. This paper presents a first-principles account by casting diverse learning paradigms in a single variational framework. We posit learning Lagrangians and show that algorithms arise as stationary points of their action, thereby providing a unifying perspective to parameter estimation tasks—covering supervised learning and generative modeling—and reinforcement learning. Table 1 provides a summary of the paper's main result. Motivated by physical principles, we postulate the corresponding learning analogy and illustrate its use in suitable learning tasks. By seeking stationary paths of the associated action, we recover classical algorithms.

**Related Work.** Machine learning and physics have early origins from energy-based models (Hinton, 2025; Hopfield, 1982) to their statistical mechanical analysis of memory capacity (Gardner & Derrida, 1988). Kaplan et al. (2020) show physics-like scaling law emerges as the neural models scale; and recent efforts have begun to analyze this phenomenon using statistical mechanics tools (Cui et al., 2021; Sorscher et al., 2022; Defilippis et al., 2024; Bahri et al., 2024; Paquette et al., 2024). Bahri et al. (2020) give a more recent survey focused on deep models. This paper, on the other hand, studies the relationship between efficient learning and the physics Lagrangian without discussing the choice of model architectures. This work derives algorithms through seeking stationary trajectories, and the commonality shared between different learning paradigms offers a unifying perspective.

Organization of the paper:

- Section 2 formalizes the connection with kinematic quantities (distance, velocity, acceleration) with Shannon information, deriving the corresponding information-processing velocity and acceleration. Insight No.1 shows that learning is a decelerating process.

- Section 3 reviews the relevant physical principles and presents the postulated learning *Lagrangians*. Solving for stationary trajectories of the associated action recovers classical algorithms in parametric models (Sec. 3.1), reinforcement learning (Sec.3.2), and parameter estimation tasks (including supervised learning and generative models)(Sec. 3.3), thereby offering a unifying perspective across seemingly disparate learning paradigms. We thus hypothesize that learning obeys the Principle of Least Action: searching for stationary paths yields learning algorithms.

## 2 LEARNING AS A DECELERATION PROCESS.

Learning in intelligent systems travels distance not in terms of space but information observed. A data stream until time $t$ is $s_1, s_2, \ldots, s_t$, abbreviated as $s_{\leq t}$. In physics, speed is defined as the rate of change of position with respect to time: $v = \lim_{\Delta t \to 0} \frac{\Delta s}{\Delta t} = \frac{ds}{dt}$. In information processing, we define position as the amount of Shannon information (Shannon, 1948) up until time t: $I(s_{\leq t}) := \log \frac{1}{p(s_{\leq t})} = -\log p(s_{\leq t})$. The rate of change of information content with respect to time, termed as instantaneous velocity in information, is thus derivable as: $v = \lim_{\Delta t \to 0} \frac{I(s_{\leq t+\Delta t}) - I(s_{\leq t})}{\Delta t}$.

In discrete information flows (e.g., language tokens) when $\Delta t = 1$, given a data stream $x_{\leq t}$, the velocity at time $t$ is $v(t) = -\log p(x_t \mid x_{<t})$. Next token prediction is thus modeling the instantaneous rate of change in information, or instantaneous velocity in information.

To check the consistency between distance and velocity in information processing, we expect it to satisfy basic physics properties, e.g., distance as an integral over velocities.

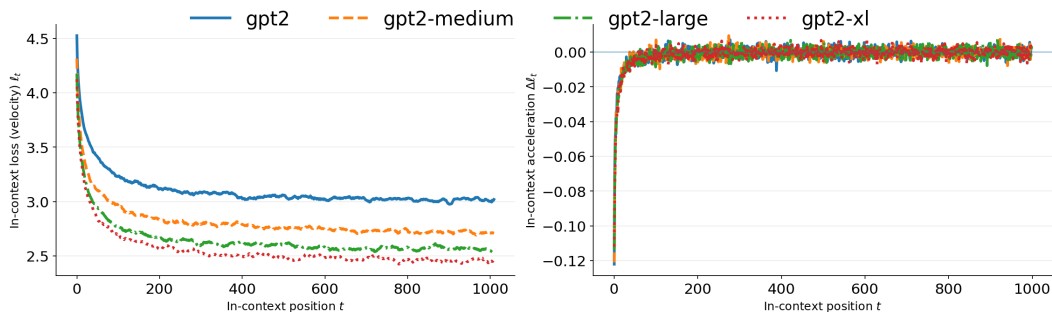

Figure 1: Expected test-time in-context learning velocity and acceleration: (Left) In-context per-token loss $\ell_t = v(t) = \mathbb{E}[-\log p_\theta(x_t \mid x_{<t})]$; (Right) In-context per-token difference in loss $\Delta \ell_t = a(t) = \mathbb{E}[\ell_{t+1} - \ell_t]$. In-context learning (as shown in the right) is a deceleration process, meaning loss goes down but less quickly as time progresses. A similar phenomenon is expected in training and test loss. Here, in-context loss is evaluated on OpenWebText.

**distance as integral.** In discrete time, physical distance satisfies: distance $= \sum_i v(t_i)\Delta t$. That holds true in information processing too: the total amount of information is the sum of chain-ruled conditional probabilities: $I(x_{\leq t}) = -\log p(x_1, \ldots, x_t) = \sum_{i=1}^t v(t_i) = -\sum_{i=1}^t \log p(x_i \mid x_{<i})$.

Continuing from understanding kinematic quantities in information processing, acceleration is the instantaneous change in velocity, defined as $a = \frac{dv}{dt} = \lim_{\Delta t \to 0} \frac{\Delta v}{\Delta t}$.

**acceleration.** In discrete information flows, acceleration models the instantaneous change in conditional probability in information processing:

$$a(t) = -\log p(x_{t+2} \mid x_{\leq t+1}) + \log p(x_{t+1} \mid x_{\leq t}) \tag{1}$$

Modelling information processing as kinematics, i.e., movements in physical spaces, prepares to understand the later postulation that learning is searching for stationary trajectories of the action. As trajectories often imply movements in physical space, here we mean movements in information space in the above sense. Considering loss curves, regardless of in-context, train, or test losses, from a kinematics perspective, provides insight No.1. Figure 1 plots the per-token in-context loss and its discrete first and second differences for small language models, corresponding to the expected test-time in-context learning velocity and acceleration.

> **Insight No.1** (Learning as a deceleration process: there is a limit $\inf v(t)$.)
>
> Generalization error on the test dataset measuring learning progress is bounded below by 0 or $\epsilon$ determined by *intrinsic* uncertainty in data. In-context loss curve, $v_\theta(t) = -\mathbb{E}[\log p_\theta(x_t \mid x_{<t})]$, $v_\theta(t)$ is a generally non-increasing function, and thus a generally decelerating process.[a]
>
> ---
> [a]Due to the monotone convergence theorem, a bounded below, non-increasing function converges to some limit. We thus hypothesize that learning converges to its infimum.

## 3   LEARNING LAGRANGIANS

Chollet (2019) measures intelligence centered around efficiency and generality, namely, when facing new tasks, an intelligent agent should adapt and acquire new skills efficiently. This idea has evolved to community challenges established in ARC-AGI-1, and ARC-AGI-2 (Chollet et al., 2025). The authors believe that intelligence is obtained through efficient learning. This paper is motivated to study the design of an efficient learning system. We present our main postulation below. We first provide a short review of relevant principles in physics and then present the corresponding learning Lagrangians. We then show that searching for the stationary path in the Lagrangians, we recover classic algorithms in different tasks.

> **Main Postulation** (Learning-by-Stationarity)
>
> Learning is searching for the path that makes action governed by the Learning *Lagrangian* stationary. In particular, learning algorithms (as in equations of motion) are obtained by seeking stationary trajectories.

**Review of Principles in Physics.**

- Fermat's Principle / Principle of Least Time (Optics) (Born & Wolf, 2019)

  A ray of light travelling from point $A$ to point $B$ chooses a path along which the time taken is the least or minimum [1]. Mathematically,

  $$T = \min_s \int_{\text{path}} n \, ds, \tag{2}$$

  where $n = \frac{1}{v}$ is refractive index and $v$ is the velocity of light in the medium.

- Hamilton's Principle / Principle of Least Action (Mechanics) (Hamilton, 1834)

  The Law states that the actual path $\xi(t)$ taken by a particle is the path that makes the action $S$ stationary, where

  $$S[\xi] = \int L \, dt = \int T - V \, dt, \tag{3}$$

  where $L$ is the *Lagrangian*, with $T$ kinetic energy and $V$ potential energy. $\xi$ is the generalized coordinates that specify the configuration of the system.

  A classic example is the Newtonian mechanics for a particle, where $\xi$ is the coordinates of the particle in the system. The Lagrangian is $L = \frac{1}{2} m |\dot{\mathbf{x}}|^2 - V(\mathbf{x}, t)$. Finding the path that makes the action stationary leads to Euler-Lagrangian equation, which gives the equation of motion $m\ddot{\mathbf{x}} = -\nabla V = F$.

- *Hamiltonian* system. The Hamiltonian system is the Legendre transform of the Lagrangian:

  $$H(\mathbf{x}, \mathbf{p}) = \mathbf{p} \cdot \dot{\mathbf{x}} - L(\mathbf{x}, \dot{\mathbf{x}}), \tag{4}$$

  where $\mathbf{p} = \frac{\partial L}{\partial \dot{\mathbf{x}}}$ is the conjugate momentum of $\mathbf{x}$.

Efficient learning is as if designing a physical system's process of walking along the information path such that it takes the least time to reach the desired error threshold. To make the idea concrete:

In learning, we define a point in space as the generalization error $\epsilon$ after observing a data sequence $\mathbf{s} := \{s_1, s_2, \ldots\}$. Efficient learning thus means optimizing for a path to reach an error threshold in the shortest time (cf. Fermat's principle of least time). Mathematically,

$$T(\delta) = \min_{\mathbf{s}} \int_0^\infty \Theta(\epsilon[\mathbf{s}] - \delta) dt = \min_{\mathbf{s}} \int_{\epsilon[\emptyset]}^{\delta} \frac{d\epsilon}{r(\epsilon, \mathbf{s})}, \tag{5}$$

where $\epsilon[\mathbf{s}]$ is the generalization error after seeing data path $\mathbf{s}$ and $\epsilon[\emptyset]$ denotes the generalization error before seeing any data, and $\Theta$ is an indicator function where $\Theta(x) = 0$, if $x \leq 0$ and $1$ if $x > 0$. Learning velocity[2], denoted by $r(\epsilon, \mathbf{s})$, is the rate of difference in generalization error as information progresses, i.e., $r_\theta(\epsilon, \mathbf{s}_n) = \epsilon_\theta(\mathbf{s}_{n-1}) - \epsilon_\theta(\mathbf{s}_n)$, where the small $\theta$ denotes the configuration of the system[3]. The least time is quantified as the least number of observations, assuming similar information content in each observation[4]. Thus we propose metrics for evaluation for efficient learning:

- sample-efficient: $T_{sample}$ = number of samples required to achieve the error threshold.

---

[1] More generally, a ray of light travelling from point $A$ to point $B$ choose an optical path that is stationary (i.e., maximum, minimum, extremum), mathematically $T = \int_A^B dt = $ stationary.

[2] We note that different learning problems with different algorithms have different rates of learning. It is derivable given specific setup and algorithm, though not known a priori.

[3] Configuration includes but not limited to model parameters, initialization, architecture choice.

[4] Future work can investigate how to quantify time when samples do not contain similar information content.

- compute-efficient: $T_{compute}$ = computational time taken to achieve the error threshold.

The metrics are proposed based on the learning time of the system indicated from Eq. 5 and the time in real life to process learning (e.g., parallel processing decreases computational time but does not enable sample-efficient learning). The above makes clear that efficient learning that could increase intrinsic intelligence requires optimization in $T_{\text{sample}}$, and investing in compute only may not be the best solution.

A natural next step is to optimize the given objective. However, we face the technical difficulty of unknown generalization error. The generalization error is derivable given a specific setup and algorithm, but it is not known a priori for optimization.

**To address the technical difficulty in optimization with unknown generalization error, we consider the following approaches:**

- *Parametric assumption.* Section 3.1 provides a concrete example in linear regression with parametric assumptions. Under suitable assumptions on input standardization, optimizing the Lagrangian given by Fermat's principle Eq. 5 yields an analytical optimal solution.

  *Remark.* Though it is not desirable in practice to constrain model classes with parametric restriction due to model mis-specification, we find it helpful to have an analytical analysis that illustrates some properties for efficient learning (e.g., planning is important).

- *Reward Hypothesis.* Section 3.2 provides insights on how reinforcement learning circumvents the problem with step-wise progress measured by reward. Writing the Lagrangian in terms of reward gives an equivalent form of Hamiltonian system, and finding the stationary path in the Lagrangian gives rise to Bellman's optimality equations (Bellman, 1958).

  *Remark.* Given the reward assumption, we will see in the section the derivation does not give rise to concrete *Lagrangian* as $L$ in Eq. 4 is replaced with reward.

- *Postulated Lagrangian.* Section 3.3 presents our postulated learning *Lagrangian* in terms of parameter estimation tasks, covering supervised learning and generative modelling. Operationalizing the learning dynamics of loss field through particle dynamics of the configuration gives rise to $\dot{\theta} = F^{-1/2}\nabla_\theta\ell$ that Adam (Kingma, 2014) approximates with diagonalized Fisher for parallel processing.

### 3.1 PARAMETRIC ASSUMPTION GIVES ANALYTICAL PATH DERIVATION.

Consider a linear regression setup: Suppose $y = x^T\beta + \epsilon$ and $x \in \mathbb{R}^p$ and $\epsilon$ has mean 0 and variance $\sigma^2$. The generalization error on the standard linear regression is:

$$\epsilon(\mathbf{x}) = \sigma^2 + \sigma^2\text{tr}((X^T X)^{-1}\mathbb{E}[xx^T]),$$

where $x$ is the test data point and $\mathbf{x}$ are the sequence of observational points as rows in the data matrix $X$. Assuming unit norm assumptions where each observed data point satisfies $||x_i||_2 = 1, \forall i$ and $x$ is uniformly drawn from the unit sphere $\mathbb{S}^{p-1}$. We work in the classical regime where $n \geq p$, so that the data matrix $X^T X$ is invertible and has full rank. Note, by unit norm assumption,

$$\text{tr}(X^T X) = \text{tr}(\sum_i x_i x_i^T) = \sum_i \text{tr}(x_i x_i^T) = \sum_i ||x_i||_2^2 = n. \quad (6)$$

Further $\mathbb{E}[xx^T] = \frac{1}{p}I_p$ due to uniform sampling over $\mathbb{S}^{p-1}$. Optimizing the Lagrangian shown in Eq. 5, we would like to choose the observational data path $\mathbf{x}$ such that $\epsilon(\mathbf{x})$ is minimized with the least number of observations. Since $S := X^T X$ is a real symmetric matrix, by the spectral theorem, there exists an orthogonal $Q$ and a real diagonal matrix $\Lambda$ such that $S = Q\Lambda Q^T$. Then $S^{-1} = Q\Lambda^{-1}Q^T$ and $\text{tr}(S^{-1}) = \text{tr}(\Lambda^{-1}Q^T Q) = \sum_i \frac{1}{\lambda_i}$. The problem of optimizing the data path:

$$\min_{\mathbf{x}:||x_i||_2=1} \int_0^\infty \Theta(\epsilon(\mathbf{x}) - \delta)dt \quad (7)$$

translates to $\min \frac{1}{p}\sum_{i=1}^p \frac{1}{\lambda_i}$ subject to $\sum_{i=1}^p \lambda_i = n$. By convexity function $t \to \frac{1}{t}$ and Jensen's inequality, one has

$$\frac{1}{p}\sum_i \frac{1}{\lambda_i} \geq \frac{p}{\sum_i \lambda_i} = \frac{p}{n}$$

The inequality is achieved when $\lambda_i = \frac{n}{p}$, thus minimum is attained at $\frac{1}{p}\sum_{i=1}^p \frac{1}{\lambda_i} = \frac{p}{n}$. Then

$$\min_{\mathbf{x}} \epsilon(\mathbf{x}) = \sigma^2 + \sigma^2 \frac{p}{n}$$

As noted before in Section 2, dependent on specific problem setup, there is an irreducible generalization error ($\sigma^2$ in this case), and the generalization error ranges from $(\sigma^2, 2\sigma^2]$ due to $n \geq p$. For example, to reach $\epsilon(\mathbf{x}) = 2\sigma^2$, the minimum sample required is $p$ and $X$ could be any orthogonal matrix $Q$. To reach $\epsilon(\mathbf{x}) = 1.5\sigma^2$, the minimum sample required is $2p$ and $X = \sqrt{2}V$, where $V$ could be any (real) Stiefel matrix. The analytical example shows us that given parametric assumptions on function classes and input distribution, it is possible to choose the observation matrix most efficiently for reducing generalization error. This is a special case for A-optimality (Atkinson et al., 2007) in linear regression setting.

A natural follow-up question is whether there is a data solution path such that adding more data points always stays along the optimal path? A short answer is no as $X^T X = \sum x_i x_i^T$ and adding one single data point to maintain $S = \frac{n}{p}I_p$ implies the added point has the property $x_i x_i^T = \frac{1}{p}I_p$, which is impossible due to rank difference between $1$ and $p$. However, adding blocks of $p$ new data points is possible, planning $p$-steps ahead in this case.

> **Insight No.2**
>
> Planning is needed to learn continuously in the most efficient way.

## 3.2 REINFORCEMENT LEARNING AS STOCHASTIC APPROXIMATION.

This section builds on two insights:

- optimizing action/policy is implicitly optimizing the data path or state path in RL terms, for learning, cf. $\min_{\mathbf{s}}$ in Eq. 5.
- The reward hypothesis circumvents the problem of unknown generalization error.

In fact, searching the stationary points in the Lagrangian written from a reward perspective derives Bellman's optimality equation (Bellman, 1958), the backbone of many RL algorithms, e.g., policy iteration, value iteration (Sutton & Barto, 2018), Q-learning (Watkins & Dayan, 1992), Deep Q-learning (Mnih et al., 2013).

*Reward Hypothesis.* All goals can be represented by rewards (Sutton & Barto, 2018).

Reinforcement learning circumvents the problem of unknown generalization error through measuring step-wise progress through reward $r(\mathbf{s}, \mathbf{a})$ on its current state $\mathbf{s}$ and next action $\mathbf{a}$. In other words, the value function $V(\mathbf{s})$ is the path to maximize reward, and the optimization over $\min_{\mathbf{s}}$ is through finding the optimal policy reaching the optimal path $V_\star(\mathbf{s})$. Greydanus & Olah (2019) provides an intuitive playground on how value function can be viewed from a path perspective. Note that the exact quantification of optimal can be incorporated appropriately through designing the reward function.

Next, we demonstrate that searching for the stationary points in the Lagrangian defined in the RL setting gives commonly known learning algorithms, i.e., Bellman's optimality equation. We do not claim novelty in this derivation, as it is textbook material in classic control theory, see Pontryagin's maximum principle (Kirk, 1970), Hamilton-Jacobi-Bellman equations for the continuous case (Evans, 2010); we include it to demonstrate the support of our main postulation that learning is searching for stationary points in the Lagrangian, and finding stationary points gives rise to classic learning algorithms.

**Derivation of Bellman equation from the Lagrangian.** The goal of the learning problem is to find actions $(\mathbf{a}_0, \mathbf{a}_1, \ldots, \mathbf{a}_{n-1})$ and states $(\mathbf{s}_0, \mathbf{s}_1, \ldots, \mathbf{s}_n)$ to maximize the objective function $J$, where

$$J = h(\mathbf{s}_n) + \int_0^{t_f} r(\mathbf{s}_t, \mathbf{a}_t, t)dt \tag{8}$$

subject to constraints $\mathbf{s}_{k+1} = f(\mathbf{s}_k, \mathbf{a}_k)$ and $t_f$ is final time. This assumes a deterministic transition where the next state is uniquely determined by its action. And $h(\mathbf{s}_n)$ is the terminal reward. Turning

the above problem into a constrained optimization problem with Lagrangians:

$$\mathcal{L}(\{\mathbf{s}\}, \{\mathbf{a}\}, \lambda) = h(\mathbf{s}_n) + \sum_{k=0}^{n-1} \left( r(\mathbf{s}_k, \mathbf{a}_k, k) + (f(\mathbf{s}_k, \mathbf{a}_k) - \mathbf{s}_{k+1})^T \lambda_{k+1} \right) \tag{9}$$

Learning a stationary solution for the Lagrangian means we search for solutions that satisfy $\frac{\partial \mathcal{L}}{\partial \mathbf{s}_k} = 0$, $\frac{\partial \mathcal{L}}{\partial \mathbf{a}_k} = 0$ for all $k$ and $\frac{\partial \mathcal{L}}{\partial \lambda} = 0$. Define discrete-time Hamiltonian:

$$H^{(k)}(\mathbf{s}, \mathbf{a}, \lambda) = r(\mathbf{s}, \mathbf{a}, k) + f(\mathbf{s}, \mathbf{a})^T \lambda \tag{10}$$

Re-writing the Lagrangian in Eq. 9 gives Eq. 11 :

$$\mathcal{L} = h(\mathbf{s}_n) - \mathbf{s}_n^T \lambda_n + \mathbf{s}_0^T \lambda_0 + \sum_{k=0}^{n-1} (H^{(k)}(\mathbf{s}_k, \mathbf{a}_k, \lambda_{k+1}) - \mathbf{s}_k^T \lambda_k) \tag{11}$$

$$d\mathcal{L} = (\nabla_{\mathbf{s}} h(\mathbf{s}_n) - \lambda_n)^T d\mathbf{s}_n + \lambda_0^T d\mathbf{s}_0 + \sum_{k=0}^{n-1} \left( \frac{\partial H^{(k)}}{\partial \mathbf{s}_k} - \lambda_k \right)^T d\mathbf{s}_k + \left( \frac{\partial H^{(k)}}{\partial \mathbf{a}_k} \right)^T d\mathbf{a}_k \tag{12}$$

With the initial position fixed ($d\mathbf{s}_0 = 0$), we search for solutions that lead to other terms of variations being 0. This leads to solutions that satisfy constraints below:

$$\lambda_n = \nabla_{\mathbf{s}} h(\mathbf{s}_n) \tag{13}$$

$$\lambda_k = \frac{\partial r(\mathbf{s}_k, \mathbf{a}_k, k)}{\partial \mathbf{s}_k} + \frac{\partial f(\mathbf{s}_k, \mathbf{a}_k)}{\partial \mathbf{s}_k}^T \lambda_{k+1} \tag{14}$$

$$\mathbf{a}_k = \arg\max_u H^{(k)}(\mathbf{s}_k, u, \lambda_{k+1}) \implies \frac{\partial H^{(k)}}{\partial \mathbf{a}_k} = 0 \tag{15}$$

Given $h(\mathbf{s}_n)$ is the terminal reward and $\lambda_n$ is the derivative of the terminal return with respect to state. That means in RL terms $\lambda_n = \nabla_s V(\mathbf{s}_n)$. Suppose $\lambda_k = \nabla_s V(\mathbf{s}_k)$. Mathematically, differentiating Eq. 16 with respect to $\mathbf{s}_k$ gives Eq. 14:

$$V(\mathbf{s}_k) = r(\mathbf{s}_k, \mathbf{a}_k, k) + V(\mathbf{s}_{k+1}) = r(\mathbf{s}_k, \mathbf{a}_k, k) + V(f(\mathbf{s}_k, \mathbf{a}_k)) \tag{16}$$

$$\nabla_{\mathbf{s}} V(\mathbf{s}_k) = \frac{\partial r(\mathbf{s}_k, \mathbf{a}_k, k)}{\partial \mathbf{s}_k} + \frac{\partial f(\mathbf{s}_k, \mathbf{a}_k)}{\partial \mathbf{s}_k}^T \nabla_{\mathbf{s}} V(\mathbf{s}_{k+1}) \tag{17}$$

Combining with Eq. 15, the solution needs to satisfy constraints:

$$V(\mathbf{s}_k) = \max_u \{r(\mathbf{s}_k, u, k) + V(f(\mathbf{s}_k, u))\} \tag{18}$$

It is not hard to see, in probabilistic transitions where the Lagrangian involves integral over randomness in the environment, the solution that satisfies being the stationary path gives:

$$V(\mathbf{s}_k) = \max_u (r(\mathbf{s}_k, u, k) + \mathbb{E}[V(S_{k+1})]) \tag{19}$$

$$u_k = \arg\max_u (r(\mathbf{s}_k, u, k) + \mathbb{E}[V(S_{k+1})]) \tag{20}$$

This is the classic Bellman optimality equation.

> **Insight No.3**
>
> The stationary path in the Lagrangian, written in terms of rewards, should satisfy Bellman's optimality equation. Thus, optimizing Bellman's equation is searching for the stationary path.

*Remark.* Recall the *Hamiltonian* system:

$$H(\mathbf{x}, \mathbf{p}) = \mathbf{p} \cdot \dot{\mathbf{x}} - L(\mathbf{x}, \dot{\mathbf{x}}), \tag{21}$$

where $\mathbf{p}$ is the conjugate momentum of $\mathbf{x}$ and $\mathbf{p} = \frac{\partial L}{\partial \dot{\mathbf{x}}}$. From the above derivation in discrete-time Hamiltonian, we saw that momentum $\mathbf{p}$ is $\lambda$ and $\dot{\mathbf{x}}$ is the transition dynamics $f(\mathbf{s}, \mathbf{a})$, and as noted

the *Lagrangian* or rate of decrease in generalization error as information progresses is replaced with step-wise reward $r(\mathbf{s}, \mathbf{a})$. Reinforcement learning thus performs well in settings with well-defined rewards, e.g., games (Mnih et al., 2015), chess (Silver et al., 2017), or verifiable problems like mathematics (Guo et al., 2025) though the lack of intermediate rewards for math problems may lead to inefficiency in search, thus large-scale training. Applying RL in real-world applications without clear rewards thus requires a carefully designed reward model, e.g., reinforcement learning from human feedback (Ouyang et al., 2022; Lambert, 2025). However, for our purposes, the above derivation does not show the learning *Lagrangian*. In the section below, we postulate the learning Lagrangian and provide reasons for our postulation.

### 3.3 Generative Models with postulated Lagrangian

In search of a design of an efficient learning system, we started from the equivalent learning Lagrangian from Fermat's Principle, to a reward-based Hamiltonian system. Efficient learning transitions from traveling on the path that takes the least time to its more general mechanical form as searching for the stationary path to minimize action.

A naïve understanding from discussions in previous sections (see Fermat's principle) would lead to the conclusion that supervised learning is less efficient than reinforcement learning, due to a lack of optimization over the data path $\mathbf{s}$. In this section, we show that this is not the case. We present our postulated Lagrangian and posit that reinforcement learning is the Legendre transform of parameter estimation tasks, in the same sense as a Hamiltonian system is the Legendre transform of the Lagrangian, such that they share the same optimal solutions.

In generative models, given a dataset $\mathcal{D} := \{\mathbf{x}_1, \mathbf{x}_2, \ldots, \mathbf{x}_n\}$, we search for parameter $\theta$ that learns how the data are distributed $p_\theta(\mathbf{x})$. Similarly, in supervised learning, we learn a conditional distribution $p_\theta(y \mid x)$ from either labelled pairs for classification tasks, or regression tasks. Both learning problems, from generative modelling to supervised learning, are parameter estimation problems.

In statistical estimation tasks, we search for an estimator $\hat{\theta}$ that maximizes the likelihood function. Here, we are not only interested in finding an estimator that best models data, but we are also looking for an efficient statistical estimator. The *Cramér-Rao lower bound* states

Let $\hat{\theta}$ be an unbiased estimator of the unknown parameter $\theta$. Then under regularity conditions,

$$\text{Var}(\hat{\theta}) - I^{-1}(\theta), \tag{22}$$

is positive semi-definite. In particular, an unbiased estimator $\hat{\theta}$ attains the lower bound, i.e., $\text{Var}(\hat{\theta}) = I^{-1}(\theta)$ is an efficient estimator. Here $I(\theta)$ is known as the Fisher information and defined as

$$I(\theta) := \mathbb{E}[(\nabla_\theta \ell(\theta; x))(\nabla_\theta \ell(\theta; x))^T] \tag{23}$$

$$= -\mathbb{E}\big[\frac{\partial^2}{\partial\theta\partial\theta^T}\ell(\theta; x)\big] \tag{24}$$

where $\ell(\theta; x)$ is the log-likelihood function. From hereon, we state the postulation.

*Postulation*: Consider the loss function $\ell(\theta, t)$ as a field[5] defined at every point of configuration $(\theta, t)$. The dynamics of the field is governed by the Lagrangian dynamics:

$$S = \int_t dt \int_\theta d\theta \int_x p(x)dx \mathcal{L}(\ell, \frac{\partial\ell}{\partial t}, \frac{\partial\ell}{\partial\theta}, \theta, t) \tag{25}$$

The integral over $x$ is due to batched sampling over data. Given the loss function in current machine learning paradigm does not depend on time, and knowing potential energy is a static term corresponding to some intrinsic property of the estimation task, we postulate it to be some log-likelihood function $\ell(\theta; x)$; knowing kinetic energy takes a quadratic form and taking into account searching for an efficient estimator, we thus hypothesize that *Lagrangian* takes the form of:

$$\mathcal{L}(\ell, \nabla_\theta \ell) = T - V = \frac{1}{2P}(\nabla_\theta \ell)^T F(\theta)^{-1}(\nabla_\theta \ell) - \ell(\theta; x) \tag{26}$$

---

[5]Here we meant by physical field.

where $P$ is the number of model parameters, i.e., $\theta \in \mathbb{R}^P$ and $F$ denotes Fisher information. Given the postulated Lagrangian, we expect the solution at the stationary points to satisfy the *Euler-Lagrangian* equation for scalar field theory with expectation adjusted:

$$\mathbb{E}[\frac{\partial \mathcal{L}}{\partial \ell}] = \mathbb{E}\Big[\frac{\partial}{\partial t}(\frac{\partial \mathcal{L}}{\partial \dot{\ell}}) + \sum_i \frac{\partial}{\partial \theta_i} \frac{\partial \mathcal{L}}{\partial (\partial \ell / \partial \theta_i)}\Big] \tag{27}$$

The left-hand side is $-1$ and due to $\mathcal{L}$ has no $\dot{\ell}$ term, the first term in the right-hand side is $0$. The second term in the right-hand side can be re-written as $\mathbb{E}[\nabla_\theta \cdot \frac{\partial \mathcal{L}}{\partial \nabla_\theta l}]$. Thus,

$$-1 = \mathbb{E}[\nabla_\theta \cdot \frac{\partial \mathcal{L}}{\partial \nabla_\theta l}] \tag{28}$$

$$-1 = \frac{1}{P}\mathbb{E}[\nabla_\theta \cdot (F^{-1}\nabla_\theta l)] \quad \text{due to } \frac{\partial \mathcal{L}}{\partial \nabla_\theta \ell} = F^{-1}\nabla_\theta \ell \tag{29}$$

Note that the divergence of a vector is the trace of the gradient of the vector. Note the Fisher does not depend on the randomness of $x$ as it already takes expectation over $x$, we have:

$$-1 = \frac{1}{P}\text{tr}(\nabla_\theta(F(\theta)^{-1}) \underbrace{\mathbb{E}[\nabla_\theta l]}_{=0 \text{ at stationary points}} + F^{-1}\mathbb{E}[\nabla_\theta^2 l]]) = \frac{1}{P}\text{tr}\big(F^{-1}\underbrace{\mathbb{E}[\nabla_\theta^2 l]}_{=-F}\big) = -1 \tag{30}$$

We thus observe (unsurprisingly) that the solution at stationary points for the parameter estimation task needs to be a maximum likelihood estimator.

The learning dynamics of loss fields needs to be operationalized through changes in particle dynamics where each parameter in the configuration $\theta$ is governed by $L = T - V = \frac{1}{2}m\dot{\theta}^T\dot{\theta} - V(\theta, t)$. Re-writing the postulated *Lagrangian*, we have $\dot{\theta} = F^{-1/2}\nabla_\theta l$ where the mass of the system is the inverse number of model parameters $m = \frac{1}{P}$ and $F$ is a symmetric and positive semi-definite matrix. In optimization, given unknown observed Fisher, we approximate using the empirical Fisher. Both RMSprop (Tieleman, 2012) and Adam (Kingma, 2014) have update based on $F^{-1/2}\nabla_\theta\ell$:

$$\textit{RMSprop: } \theta_{t+1} \leftarrow \theta_t - \alpha\frac{g_t}{\sqrt{v_t} + \epsilon}, \tag{31}$$

$$\textit{Adam: } \theta_{t+1} \leftarrow \theta_t - \alpha\frac{\hat{m}_t}{\sqrt{\hat{v}_t} + \epsilon}, \tag{32}$$

where $g_t = \nabla_{\theta_t}\ell$, $v_t = \beta_2 v_{t-1} + (1 - \beta_2)g_t \odot g_t$, and $m_t = \beta_1 m_{t-1} + (1 - \beta_1)g_t$, $\hat{m}_t = \frac{m_t}{1-\beta_1^t}$, $\hat{v}_t = \frac{v_t}{1-\beta_2^t}$, and $\epsilon$ are added for numerical stability. From the Lagrangian, one can also predict the inefficiency of SGD, as it does not satisfy the Euler-Lagrange equation. Combining with Section 3.2 on the relationship with reinforcement learning and Hamiltonian system, we thus posit our insight:

> **Insight No.4**
>
> Reinforcement learning is the Legendre transform of parameter estimation tasks under Adam / RMSprop optimization.

## 4 CONCLUSION

Motivated by the study of efficient learning through physics, we find surprising synergies between different physics principles and different learning paradigms, from active data selection, reinforcement learning, to parameter estimation tasks. We assay the results in Section 3 and derive classic learning algorithms from seeking stationary trajectories in the *Lagrangian*, offering a unifying perspective to seemingly broad and different learning paradigms. As any intriguing hypothesis needs experimental verification, a natural next step is to design verifiable experiments. Though at the current status, we find our insights with mathematical justification provide a diverse range of postulations about synergies across different fields that could require community efforts to test and verify.

ETHICS STATEMENT

The paper aims to understand the fundamentals of learning and intelligence. We demonstrate a close connection between physics and learning and postulate that learning, too, follows physical laws. This work promotes the importance of AI safety and ethics, as machine learning, like other engines or entities, obeys the laws of Nature. This paper presents a principled, promising approach to designing safer AI through understanding the fundamental laws behind learning.

REPRODUCIBILITY STATEMENT

The paper includes theoretical derivations within the paper and experiment results are easily reproducible through public sources.

THE USE OF LARGE LANGUAGE MODELS

Large language models are used to polish academic writing, search for references, and provide hints for mathematical proofs with concrete prompts. Large language models are very helpful as an assisted tool, but it still cannot directly contribute to the paper's main contribution.

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
