# OpenReview forum: "Physics of Learning: A Lagrangian perspective to different learning paradigms"
_ICLR.cc/2026/Conference — Submitted to ICLR 2026_

### Official Review · Reviewer_N1Zv · 2025-10-24

**Soundness:** 2
**Presentation:** 2
**Contribution:** 2
**Rating:** 2
**Confidence:** 2

**Summary:**

The paper expresses various learning setups through variational functionals. The authors follow a well-known formalism from physics (Lagrange functions/principle of least action). The examples include active learning, the Bellman optimality equation, and supervised learning.

**Strengths:**

1) The paper studies analogies to variational calculus in physics/math. This could allow to bring well-established techniques from other fields to machine learning research.

2) The authors attempt to unify different learning paradigms, which could create a deeper understanding of learning mechanisms.

**Weaknesses:**

1) It is difficult to read the paper.

- Generally, it would help to introduce the symbols in a more mathematical way (discrete/continuous, dimension).
- At the beginning (L209,L216), sample-efficiency and compute time are introduced as a characterization of efficient learning but the latter seems to play no role in the paper. Omitting needless concepts could make it easier to read.
- The purpose of Section 2 and Insight 1 is not clear to me. It appears to be a side observation that is not relevant for the core topic of the paper.
- The standard Lagrangian/Hamiltonian frameworks, as I know them, usually starts with a clear declaration of the variables and their conjugate variables. I would recommend this for each of the examples.
- Insight 2: Planning usually refers to using a learned/known model of the environment for better action selection.

2) The contribution to derive learning algorithms from Lagrangians is not as strong as claimed.

- Section 3.1 appears to me to be specific to linear regression.
- The formulation of the Bellman equation through a variational principle is (as the authors say) not novel.
- It is claimed that the Adam optimizer is derived. However, it looks more like a derivation of maximum likelihood estimation in Section 3.3 and that Adam is one way of approximating this. I doubt that Adam with all its features, such as normalization of update sizes, can be derived from a simple Lagrangian like the one in Equation 26.

**Questions:**

1) Time evolution in physics is unique and so is the solution of the corresponding variational formulations. How is this uniqueness interpreted in a learning context? Is there a unique best learning algorithm?

2) How can a formulation in terms of variational principles benefit machine learning research? In math/physics, it allows to incorporate symmetries more easily, using certain proof techniques, geometric invariance, ... Can you demonstrate any of these advantages?

---

> ### Author Response · Authors · 2025-11-24
>
> We thank the reviewer for their constructive feedback and especially interesting questions. We will address the concerns below.
>
> **(L209,L216), sample-efficiency and compute time.** We thank the reviewer for pointing out this detail. This is actually deliberate, though we agree we didn’t have the space to explain in more detail. What we found interesting is that the equation elucidates the importance of sample-efficiency. Often in ML community, we talk about efficient learning in a liberal format, both in terms of computational efficiency through parallelization and sample efficiency. In this paper, we would like to emphasize that sample-efficiency is an important metric for intrinsic intelligence.
>
> To see this concretely, consider a thought experiment: Human A and Machine B process 10 samples and reach the same learning capability, measured by generalization error. Human A takes 10 seconds to process, Machine B takes 0.1 seconds to process. That means in real-life processing time, Machine B will process 100x samples in 10 seconds and thus more intelligent. The intelligence is driven purely by the speed of processing. But if Machine B can reach the same generalization error with only 2 samples, that is intrinsic intelligence, learning with fewer samples. A concrete future work is to test whether current models are achieving super-level intelligence through compute efficiency or both sample efficiency too, and what are algorithms can drive improvements on both front. We will make this point clearer in the next version.
>
> **Purpose of Section 2 and Insight 1.** We thank the reviewer for the concern. The purpose of Section 2 as illustrating the movement on information space when designing an efficient system. This echos with Section 3.1 with movement in generalization error. We agree it is not the main point of the paper but we deem it as preparation material to understand what is meant by kinematic movement in information space. We will make this point clearer in the next version.
>
> **clear declaration of the variables, introduce the symbols.** We thank the reviewer for the suggestion and will incorporate them into the next version.
>
> **Insight 2 Planning.** We thank the reviewer for this interesting observation. As demonstrated in Section 3.2, optimizing in action is implicitly optimizing state spath, as next state is determined by current state and action choice. Thus planning over state and planning over action governed by transitional dynamics is the same thing. Part of the paper's scientific contribution is through a unifying lens to illustrate the hidden connections more clearly.
>
> **Contribution in deriving learning Lagrangians.** We thank the reviewer for the concern and appreciate the reviewer’s good understanding on technical details. However we hope the reviewers do see re-derivation of classical learning algorithms is the first step through the lens of Lagrangian. The main contribution is to demonstrate that learning in many paradigms follows the a unified least action principle. We wonder *whether learning, itself, behaves like a physical process governed by a unified action principle.*

---

> > ### Author Response · Authors · 2025-11-24
> >
> > We thank the reviewer for the very deep and insightful questions, without proper investigation we could only offer a dicussion with the reviewer.
> >
> > **Time evolution in physics.** So far the learning algorithms are unique solution from the derivation. However if we observe current machine learning practices, we are combining many of those learning algorithms ad-hoc to achieve good performance. For example, deep RL is a field combining control theory and parameter estimation tasks. Therefore we think composition of learning algorithms may be very interesting direction to study. We would be keen to hear how reviewer views the significance of unique solution in learning systems.
> >
> > **Benefits in variational principles formulation on machine learning research.**
> > The authors see the potential next steps for this work as three-fold:
> >
> > First, it opens the community with a starting point on the investigation of a principled way to derive learning rules beyond existing paradigms, like active learning, and unsupervised learning.
> >
> > Second, it offers a language for analyzing learning dynamics using tools familiar to physicists—action functionals, symmetries, conservation-like quantities—opening the door to connections with physics and AI. (as the reviewer insightfully also pointed out in the question)
> >
> > Third, it generates concrete, testable predictions about optimization behavior and sample efficiency (e.g., when and why certain optimizers outperform others with consideration of designing kinetic energy terms), which can guide both algorithm design and empirical study in modern deep learning. We also agree with the reviewer in its current simplest form Adam with all its features are not derivable. But the interesting point is here would be a place to start incorporating design choices with links to physics to aid understanding and see if it generates predictable effects in experiments.
> >
> > The paper in its current form’s contribution is mainly conceptual. But it may be already intriguing for both theoretically driven and methodology focused ML audience. There are many natural next steps suggested by this framework, several of which we see as important directions for future work but are challenging to fully address within the scope of a single paper, given the rather novel nature of the theory with no prior well-established work to build on. We would therefore very much welcome further discussion and feedback from the community.

---

### Official Review · Reviewer_hTkZ · 2025-11-01

**Soundness:** 4
**Presentation:** 3
**Contribution:** 2
**Rating:** 6
**Confidence:** 3

**Summary:**

This paper proposes interpreting efficient learning as a physical process that makes a Lagrangian stationary. The authors present concrete examples with mathematical derivations, including linear regression, the Bellman equation in reinforcement learning, and Adam/RMSProp update rules.

**Strengths:**

- This paper is well-written and easy to follow. The structure is clear, and the explanations are concise yet informative.
- The unifying viewpoint of learning algorithms as physical processes is intriguing.

**Weaknesses:**

I am concerned about the practical implications of this framework. While the perspective is theoretically appealing, it is not clear how it leads to tangible benefits for designing or improving learning algorithms in practice. For example, some prior work applies physical concepts to optimization and reports practical improvements (see, e.g., https://arxiv.org/abs/1906.01563).

**Questions:**

Does the current framework have the potential to inspire new algorithms or learning paradigms? Even without experiments, a discussion of possible practical uses would strengthen the paper's impact.

---

> ### Author Response · Authors · 2025-11-24
>
> We thank the reviewer for the constructive feedback and appreciation on the novelty of the work.
>
> **Practical implications and re Questions.**
>
> We also thank the reviewer for pointing out the interesting reference and will incorporate a discussion in the next version. For example, the prior work studies incorporating inductive biases in learning systems to enable learning the laws of physics; whereas this work studies *whether learning, itself, behaves like a physical process governed by a unified action principle, including demonstration on control theory, active learning and parameter estimation tasks.*
>
> The authors see the potential next steps for this work as three-fold:
>
> First, it opens the community with a starting point on the investigation of a principled way to derive learning rules beyond existing paradigms, like active learning, and unsupervised learning.
>
> Second, it offers a language for analyzing learning dynamics using tools familiar to physicists—action functionals, symmetries, conservation-like quantities—opening the door to connections with physics and AI.
>
> Third, through investigating the Lagrangian dynamics, it could generate concrete, testable predictions about optimization behavior and sample efficiency (e.g., when and why certain optimizers outperform others with consideration of designing kinetic energy terms), which can guide both algorithm design and empirical study in modern deep learning.
>
> In conclusion, the authors think the paper in its current form may be conceptually intriguing for both theoretically driven and methodology focused ML audience. There are many natural next steps suggested by this framework, several of which we see as important directions for future work but are challenging to fully address within the scope of a single paper, given the novel nature of the approach. We would therefore very much welcome further discussion and feedback from the community.

---

### Official Review · Reviewer_2i1D · 2025-11-04

**Soundness:** 3
**Presentation:** 2
**Contribution:** 2
**Rating:** 2
**Confidence:** 5

**Summary:**

The paper identifies a connection between learning and mechanical systems based on the principle of least action. The paper shows how certain well known machine learning algorithms can be derived by considering the Lagrangian.The algorithms span different types of learning such as parametric models, reinforcement learning and generative models.

**Strengths:**

The paper makes a connection between physical systems and learning ones. It does this using a Lagrangian and the principle of least action. The idea is that a sample efficient learning process should correspond to a stationary trajectory. Rather than in physical space and indexed by time, this is in information space indexed by sample number. The similarity of the two types of processes is striking, and it's nice to see a conceptual paper. Using this, the authors give derivations of three types of learning algorithm: parametric, reinforcement and supervised learning. The fact the types are somewhat different lends strength to the main argument of the paper. However, there is a fatal flaw.. see Weaknesses.

**Weaknesses:**

The analogy--central to the paper--between learning algorithms (which are optimal ways of accruing some kind of quantity) and solutions to Euler-Lagrange equations is not totally surprising. And what's worse, this observation has even been explicitly made before, see for example a blogpost by John Baez about the Bellman Equation from 2014: https://johncarlosbaez.wordpress.com/2014/10/30/sensing-and-acting-under-information-constraints/ which summarizes some points from a talk by Naftali Tishby. Tishby is not cited in the present paper.

I think the paper should better explain what parts are novel and what have been done before, showing clear novel content,  in order to warrant consideration in a venue like ICLR. The presentation in the paper would also need to be significantly improved, though the presentation issues are a moot point given the problem with the central premise. For future work, here are some presentation issues:

The paper has an idiosyncratic style. For example, immediately after the abstract, a table is used to explain some the main ideas. Moving the table would help somewhat, but I suggest a more textual introduction would better help readers understand.

Sloppy wording regarding physics is also a recurring problem in the paper. This even starts in Table 1 (the first thing the reader comes across!) with phrases such as:  *" .. from first principles, i.e., the Learning Lagrangian"*, though the Learning Lagrangian itself is not a principle; or *".. searches for stationary paths in the Lagrangian"* where the authors presumably mean this as shorthand for ".. searches for paths with stationary action, where action is defined in terms of the Learning Lagrangian". Freewheeling wording like this is especially disconcerting in a paper whose main contribution is revealing an analogy between phenomena in different areas.. it's important for the technical statements of these phenomena to be stated rigorously, or at least following conventional shorthand.

The introduction section, itself, is rather short even though it includes a somewhat terse related work section, typically a section in its own right. I suggest it should be separated and expanded. Throughout the paper, various ideas are introduced using words or sections in bold print rather than a more traditional structure.

Within the paper there are a few mentions of limitations, or assumptions, it would be nice to collect these and have a limitations and future work section before or part of the conclusions.

There is a lack of experimental validation which the authors themselves recognize. Depending on the venue where this is submitted, that might not be an issue (e.g. in a book chapter), but especially for a machine learning conference, and given that the central analogy is not totally novel, it would be good to have at least some practical benefit from the ideas.

**Questions:**

Is Table 1 really intended to have the words "Fermat's principle, Hamiltonian, the Lagrangian" in the bottom left cell? The lists of Applications and of Algorithms in the bottom right and center cells seem like lists of what can be obtained as stationary solutions under a corresponding  Learning Lagrangian, and the applications in which these occur respectively. But what's the bottom left cell doing? And presumably there are actually two tables one on top of the other, or are Applications really meant to be under Physics, and Algorithms under Learning?

---

> ### Author Response · Authors · 2025-11-24
>
> We thank the reviewer for taking the time to provide feedbacks. The concerns mainly centered around novelty and presentation clarity, we here will mainly address the novelty concern.
>
> **On Novelty.** We thank the reviewer for pointing out the very interesting work on Tishby, we will include citation in the next version. However, we note throughout the paper, starting from the front page in the Table 1 we used [†] to denote is classic textbook material in control theory. The contribution of the work is even though classic control theory gives reward as replacement for Lagrangian, which is unknown from the observer’s perspective as it is subjective and given by environment. This work still searches for a Lagrangian dynamic governed by kinetic energy and potential energy and proposed a formulation in Section 3.3, with initial mathematical derivation. So novelty contribution is on postulation in Section 3.3, and the connection to active learning in Section 3.1 as indicated very early on in the paper through [*]. Those investigation support our central research question: *whether learning, itself, behaves like a physical process governed by a unified action principle, including control theory, active learning and parameter estimation tasks.*
>
> **On presentation.** We thank the reviewer for the suggestions on presentation formats and would incorporate accordingly in the next version.
>
> **Lack of experimental validation.**  We thank the reviewer for raising this concern. While we agree that experimental validation is important, our work is motivated precisely by the fact that current techniques and standard frameworks have not led to a satisfactory understanding of intelligence. This, in our view, underscores the importance of exploring unconventional theoretical frameworks as complementary attempts to understand learning. We thus hope that ML venues remain open to novel and unconventional contributions, in addition to works that focus on improvements within established paradigms. The authors think given that this paper is exploring a rather new territory, with many meaningful future directions to incorporate and validate, it might be interesting to present this conceptual framework to the wider audience.
>
> **Question on Table.** The table shows the corresponding physics objective according to each principle, where the work presents the corresponding proposed or existing learning objective. Each learning objective is applicable to each specific learning application, for example with generalization error involved in order to compute, one needs to assume parametric models; with reward function involved, one is assumed to be in reinforcement learning setting; with parameter $\theta$ involved, one is in parameter estimation task setting. And each learning objective while searching for a stationary point, gives rise to a corresponding algorithm. We hope this clarifies and happy to engage in further discussions.
>
> Since the reviewer’s concerns mainly pertain to presentation, which we consider addressable without affecting the work’s scientific contribution, we feel there may be some misunderstanding of the paper’s central message. We would kindly ask the reviewer to reconsider their assessment of the work’s novelty. In line with comments from other reviewers (reviewer hTkZ: “intriguing”; reviewer N1zv: “could create a deeper understanding of learning mechanisms”), we believe this work explores a novel theoretical territory for learning and can contribute to a deeper understanding of underlying ML mechanisms.

---

> > ### Comment · Reviewer_2i1D · 2025-11-25
> > **Reply is not in good faith**
> >
> > First, it is a serious mis-representation of my review to say my "concerns mainly pertain to presentation" (your words in final paragraph of your reply). As I made clear in my summary of the paper, and throughout my review, there is a "fatal flaw" in the paper, namely the fact that the core idea is not novel, yet is claimed to be.
> >
> > Devoting an entire paper to elaborating on the idea of another researcher as though it were novel is potentially plagiarism, though I am willing to believe the Authors here were simply unaware of that background work. However, even if that work were to suddenly be cited (as the Authors propose), there is a further problem which is that the paper does not add applications or analysis that build on Tishby's perspective, rather it (sells that perspective as the Authors' and) just adds detail to that perspective. This could be appropriate in a survey paper or a textbook. This is not the current context. Once the core idea is properly attributed there is not enough original work left to meet the bar for an ICLR paper.
> >
> > Second, the Authors did not directly answer my questions about the table. The first question needs a **yes/no reply**. The last question needs a **yes/no reply**. In between is a specific **request for explanation about the bottom left cell**. It is not good faith communication to avoid answering these questions and instead deflect attention with a vague high-level description.
> >
> > And finally, the Authors cite two other reviewers who did not know of Tishby's work, suggesting their positive comments should help me reconsider my point about novelty, and that "there may be some misunderstanding [by me] of the paper’s central message" (!). In reality, all reviewers, including me, agree about the paper's central message:
> > - "the authors introduce a new perspective through which to view the learning algorithms" (Sh1j);
> > - "The authors propose a framework to connect and unify several learning settings through the language of classical physics, specifically the principle of least action." (Szcw);
> > - "The paper identifies a connection between learning and mechanical systems based on the principle of least action. The paper shows how certain well known machine learning algorithms can be derived by considering the Lagrangian." (2i1D);
> > - "This paper proposes interpreting efficient learning as a physical process that makes a Lagrangian stationary." (hTkZ);
> > - "The paper expresses various learning setups through variational functionals." (N1Zv)
> >
> > It is not appropriate to try to brush aside the serious issue of non-novelty of the paper's central message by mis-representing my review as focused on presentation and mis-representing the other reviews as confirming novelty of a (different) message.

---

> > > ### Author Response · Authors · 2025-11-27
> > >
> > > We thank the reviewer for their comments, which we address below.
> > >
> > > **On the alleged “fatal flaw”.** The reviewer claims there is a fatal flaw in our paper, citing a blogpost by John Baez about the Bellman Equation from 2014: https://johncarlosbaez.wordpress.com/2014/10/30/sensing-and-acting-under-information-constraints/.
> > >
> > > 1. We **do not** claim the connection between the Euler-Lagrange equation and the Bellman equation to be a novel result. In fact, we **explicitly highlighted in multiple places that it is not a novel result**. See Table 1 on Page 1 (“classic textbook material in control theory”) and also see line 312 - 316 as cited below:
> > >
> > > > “We do not claim novelty in this derivation, as it is textbook material in classic control theory, see Pontryagin’s maximum principle (Kirk, 1970), Hamilton-Jacobi-Bellman equations for the continuous case (Evans, 2010); we include it to demonstrate the support of our main postulation that learning is searching for stationary points in the Lagrangian, and finding stationary points gives rise to classic learning algorithms.”
> > >
> > > The blog post cited by the reviewer only overlaps with our work on the topic of the relationship between Bellman optimality and Euler-Lagrange – which as seen above, we explicitly claimed to be not novel, referencing earlier work.
> > >
> > > Thus, we **strongly disagree** with the reviewer’s claim that we presented the work laid out in the cited blog post (or the reference contained In the blog post) as novel: in fact, we did the polar opposite. If the reviewer disagrees, we would invite them to provide detailed evidence between the overlap with the cited blogpost and the result we actually claimed as novel in the paper.
> > >
> > > For the avoidance of doubt, the core claim of novelty in the paper is summarised below:
> > >
> > > *We hypothesize that learning, itself, behaves like a physical process governed by a unified action principle. We offered an analysis across parametric models, reinforcement learning, and generative/supervised learning can all be viewed as searching for stationary trajectories of an appropriate functional, making explicit connections. In particular, we presented two new learning Lagrangians to the best of our knowledge. They do not seem to be covered in the rest of the literature and the reviewer’s reference.*
> > >
> > > *These two learning Lagrangians **do not** include any material from the cited post by John Baez and the exposition of the third learning Lagrangian (Bellman Equation) was explicitly highlighted as not novel in the paper.*
> > >
> > > **Re direct answer about the table.**
> > >
> > > The reviewer said “it is not in good faith communication” and claimed the authors attempted to “deflect attention”.
> > >
> > > We **strongly** disagree with this characterization of our earlier reply.
> > >
> > > It is **not in our interest** and certainly **not our intention** to deflect from a simple to answer table formatting question.
> > >
> > > Hence, we apologize in case a lack of clarity in our earlier reply cause offense.
> > >
> > > As per the reviewer’s request for a single-word answers:
> > >
> > > >Question 1: Is Table 1 really intended to have the words "Fermat's principle, Hamiltonian, the Lagrangian" in the bottom left cell?
> > >
> > > Answer: **yes**.
> > >
> > > > Question 2: But what's the bottom left cell doing?
> > >
> > > Answer: Assuming the bottom left cell refers to "Fermat's principle, Hamiltonian, the Lagrangian", for the sake of readability, we present a table with three rows and four columns as a table of three rows and two columns (stacked vertically). Therefore, both tables share the same row labels, as they are simply a single four-column table split in two.
> > >
> > > > Question 3: And presumably there are actually two tables one on top of the other, or are Applications really meant to be under Physics, and Algorithms under Learning?
> > >
> > > **Yes** (to the first option); **No** (to the second option). The detailed explanation is given in the answer to question 2.
> > >
> > > We thank the reviewer for pointing out that this is not clear and will include a note to clarify the formatting in the next version of the paper.
> > >
> > > **Regarding citing other reviews.** The reviewer states “two other reviewers ... did not know of Tishby's work”. This may or may not be the case - the other reviewers did not mention knowing or not knowing Tishby’s work.
> > >
> > > We refer to point (1) and maintain that the results the reviewer criticized and cited as not novel were never claimed to be novel in the paper. Therefore, we suspected a lack of clarity in our presentation pertaining to what we claim or do not claim to be novel, despite our demonstrated best effort to be explicit in the paper (again, see (1)) -  - thus our statement about presentation.
> > >
> > > We apologize if we created the impression that we did not take the reviewer’s concern seriously. Our point is not to “brush aside” concerns or to “mischaracterise” the review: In fact, we would very much like to address it and believe that we have done so above.

---

### Official Review · Reviewer_Szcw · 2025-11-04

**Soundness:** 1
**Presentation:** 2
**Contribution:** 1
**Rating:** 2
**Confidence:** 3

**Summary:**

The authors propose a framework to connect and unify several learning settings through the language of classical physics, specifically the principle of least action. The framework describes learning as the minimisation of a Lagrangian where, in place of physical notions of position and time, the authors instead consider a dataset that is iteratively added to. They provide three applications of this framework—to experimental design, reinforcement learning, and optimisation—recovering classical machine learning results such as A-optimality, Bellman’s optimality equation, and natural gradient descent.

**Strengths:**

- The paper identifies a connection between physics and machine learning, two fields of interest to the conference.
- The authors attempt to unify concepts in experimental design, reinforcement learning, and second-order optimisation. In this sense, the work is refreshingly ambitious.
- The proposed formulation of active learning/experimental design via physical analogues appears to be a novel perspective, though I am not familiar enough with this specific literature to comment on its novelty with confidence.

**Weaknesses:**

- The paper’s clarity is a significant concern. Many explanations, especially in Section 3, feel underdeveloped or are presented too hastily. The terms “postulate”, “conjecture”, and “hypothesise” are used frequently, often immediately before a strong claim is made, which blurs the line between assumption and proof.
- The paper uses physics notation and ideas without sufficient exposition. A large proportion of the ICLR audience has a background in Mathematics and Computer Science; notation that is not standard in these fields should be clearly defined. I would encourage the authors to add a dedicated background/notation section.
- The paper's motivation is somewhat puzzling. The authors claim a "surprising link" between physics and ML, yet also describe how important physics has been to ML's development. The fact that different learning problems can be cast as optimisation problems (minimising a Lagrangian) is not suprising given the role that classical physics has played in this development.
- The contribution seems overstated on several occasions. The authors claim their work provides physical justification for the Adam optimiser, but the analysis is for natural gradient descent, which is theoretically distinct. Similarly, the claim of applications to generative models is justified by this same NGD analysis. This connection feels tenuous and appears to overstate the framework's relevance to contemporary generative models.
- "Insight No. 1" (that learning is a decelerating process) is a well-known property of learning curves and may not constitute a novel insight of this framework.
- The "Main Postulate" is conceptually challenging. It states that learning is a search for a stationary path (data sequence), but in most learning settings (outside of experimental design), the data path is fixed, and the search is over the model class.
- The paper's contribution appears thin for a main-track submission. It lacks novel theoretical guarantees or new experimental results. The primary contribution is the proposal of a framework and various conjectures. However, the framework is presented in a way that is difficult to follow, which unfortunately detracts from its value as a contribution.
- The equality on the right-hand side of (5) doesn’t appear to be correct (see questions).

**Questions:**

- Could the authors please provide a derivation for the equality on the right-hand side of equation (5)?
- Are the authors familiar with the online-to-PAC framework (Lugosi and Neu, 2023)? It might be relevant to their work.
- I found section 3.3 very difficult to follow. Could the authors please clearly explain this, taking greater care to state what things are assumed/hypothesized and which parts are mathematical deductions?

---

> ### Author Response · Authors · 2025-11-24
>
> We thank the reviewer for their time and appreciate the opportunity to clarify several points.
>
> **On novelty.**
> This work, to the best of the authors knowledge, is the first to cast learning as *stationary-action* dynamics on information fields with a unifying reinterpretation of existing learning paradigms and optimizers through that formalism. This echoes with reviewer Sh1j on “clearly an original piece of work”.
>
> **On clarity and notation.** The authors thank the reviewer for pointing out and will improve on clarity and notation in the next version.
>
> **On the role of classical physics in ML development.** We agree with the reviewer on the intertwined development between physics and machine learning: e.g., Hopfield networks, Boltzmann machines, energy-based models. In this sense, there is a long and important tradition of physics-inspired ML - and this may not be a coincidence.
>
> What we found surprising is that through derivations, the learning dynamics themselves admit a principled derivation from a single least-action postulate: the same variational formalism yields simpler form of A-optimality, Bellman’s equation, and Adam/RMSprop as stationary dynamics of carefully chosen learning Lagrangians. In other words, we wonder whether *learning, itself, behaves like a physical process governed by a unified action principle.* The work posits to study the connection more formally, such that the physics-inspired ML methodology do not need to be ad hoc but can be principally derived if a proper theory is formed.
>
> **On Adam and NGD.** We would like to clarify: we regard the analysis is for Adam as the learning dynamics is over a scalar field where the log-likelihood is scalar governed by many degrees of freedom $\theta$. The stationary point is a MLE given Fisher-based kinetic energy. To approximate the dynamics of this scalar field, we use particle dynamics, where particle are parameters of the neural network which is governed by $½ mv^2$, where velocity is the change with respect to time of a single parameter theta, that is $F^{-½} \nabla_\theta \ell$ and hence Adam. If it is NGD, then the kinetic energy would constitute $\frac{1}{2P} \nabla_\theta \ell F^{-2} \nabla_\theta \ell$, with emphasis on $-2$ exponent.
>
> **On relevance for Insight No.1.** We agree with the reviewer by itself it is not too novel. However, we would like to show the connection between information space and physical space, in terms of position = shannon’s entropy, velocity = next token prediction and similarly acceleration. This is to aid understanding of the learning system’s kinematic movement in information space when we refer to later in designing an efficient learning system as if walking along an information space.
>
> **On clarification on main postulation.**  We would like to point out the difference: as reviewer correctly pointed out that in experimental design, the learning is searching over data sequence; we also demonstrated in Section 3.2 that in reinforcement learning, the learning is searching over (state, action) pair to achieve cumulative reward. Interestingly, when optimizing action, due to next state is determined by transition dynamics of current state and current action, we are implicitly optimizing state path, which is data path in RL terms. Moreover, in Section 3.3, we especially postulated the search over model space where the Lagrangian dynamics is governed by parameter space, i.e., model class as reviewer pointed out. This supports the central insight of our paper that many *learning paradigms itself behaves like a physical process governed by a unified action principle, where algorithms are derivable.*

---

> > ### Author Response · Authors · 2025-11-24
> >
> > **Derivation on RHS of Eq. (5).** Certainly, it is done through change of variables.
> > We define $r(\epsilon, s)$ as the rate of change of $\epsilon$ under observing sequence $s$ which changing over time. So $\frac{d\epsilon}{dt} = r(\epsilon, s)$ by definition. Assuming $r(\epsilon, s) \neq 0$ and that $\epsilon$ moves smoothly, $dt = \frac{d\epsilon}{r(\epsilon, s)}$.
> > So the LHS of eq. (5) is the earliest hitting time for epsilon to reach threshold $\delta$,
> > $\int_{t:\epsilon > \delta} dt = \int_{\epsilon[\emptyset]}^\delta d\epsilon / r(\epsilon, s). $
> > Putting the minimization in gives the result.
> >
> > **On online-to-PAC framework (Lugosi and Neu, 2023).** We thank the author for the helpful pointer and would incorporate a discussion on the relevant work in the next version.
> >
> > **Explanation on Section 3.3.**
> > We assume the Lagrangian dynamic governed by log-likelihood function follows standard physical form kinetic energy - potential energy where kinetic energy is $\frac{1}{2P}\nabla_\theta \ell F^{-1} \nabla_\theta \ell $ and potential energy is $\ell$.
> > We consider the learning problem as a scalar field theory, where $\ell$ is the log-likelihood scalar and governed by many degrees of freedom $\theta$. Then the stationary point at least action principle should satisfy Euler-Lagrangian equation for scalar field theory, i.e eq. (27).
> > Upon mathematical deductions from Eq. (28 - 30), the solution that allows Euler-Lagrangian equation to hold is a MLE estimator.
> > In order to reach MLE estimator, the desired solution at stationary point, we require to approximate the learning dynamics of loss fields with particle dynamics through change in parameter space. Thus the Lagrangian dynamic for field is approximated with lagrangian dynamic for particle where change of theta with respect to time in kinetic energy corresponds to $F^{-½} \nabla_\theta \ell $ which RMSprop and Adam are based on for diagonalized Fisher.
> >
> > We are happy to engage in further discussions. What we found interesting in terms of Section 3.3, is the stationary point is an MLE estimator and to approximate the learning dynamics of field gives out RMSprop or Adam $F^{-½}$ update. A quick mental check is without Fisher conditioned kinetic energy, Euler Lagrangian for scalar field theory would not equal on both sides.

---

### Official Review · Reviewer_Sh1j · 2025-11-09

**Soundness:** 4
**Presentation:** 3
**Contribution:** 3
**Rating:** 8
**Confidence:** 4

**Summary:**

In the present paper the authors introduce a new perspective through which to view the learning algorithms driving modern machine learning. Applying Lagrangian mechanics, stationary paths are looked for to re-derive A-optimality, Bellman’s equation for reinforcement learning, and the widely used optimization algorithms Adam, and RMSprop from a dynamical systems angle. Observing the unification of these learning algorithms under the same formalism, the authors posit that learning dynamics obey the principle of least action.

**Strengths:**

The core strengths of the presented work are two-fold:
* Intellectual clarity of the presented ideas, this includes the derivation of the results, and their motivation.
* Rigor of derivations

While clearly an original piece of work, the paper builds on a large foundation of physics-based approaches in machine learning. Beyond the intellectual clarity, the presented work scores highly with regards to its significance with the potential help and understanding the result provides.

**Weaknesses:**

* At its core, the presented approach introduces a structured treatment of the space of learning dynamics, and can as such be interpreted as leveraging the information geometry [1]. The authors sadly fail to draw this connection,
* There exists a wide body of literature at the intersection of machine learning’s learning dynamics, and approaches from physics such as classical mechanics, and statistical physics as pointed out by the authors (line 72-80). Nevertheless, there exist a number of works who specifically target the learning dynamics with a variety of techniques from classical mechanics to which this paper should be set in relation to. In order of proximity to the presented work:
    * The work by Winer and Hanin [2] takes the perspective of viewing the network as a Hamiltonian over its inputs, and is hence closely related to the present work.
    *  Redman et al. [3] utilize Koopman operators to characterize the learning process, and are hence arguably orthogonal to the present work.

References:
1. Ollivier, Yann, et al. "Information-geometric optimization algorithms: A unifying picture via invariance principles." Journal of Machine Learning Research 18.18 (2017): 1-65.
2. Winer, Mike, and Boris Hanin. "Deep Neural Nets as Hamiltonians." arXiv preprint arXiv:2503.23982 (2025).
3. Redman, William, et al. "Identifying equivalent training dynamics." Advances in Neural Information Processing Systems 37 (2024): 23603-23629.

**Questions:**

* Re-deriving algorithms through the lens of Lagrangian dynamics as a latent unifying principle in itself is highly thought-provoking, while the re-derivation is one step, how do you envision to leverage the developed formalism going forward to aid in the development of new algorithms?
* Can the authors’ formalism, and posited principle of least action, be extended to more advanced forms of differentiation such as natural gradients, and the resulting algorithms such as natural gradient descent?

---

> ### Author Response · Authors · 2025-11-24
>
> We thank the reviewer for the careful reading, the pointers to interesting and relevant literature.
>
> **On the connection to information geometry.**
> We agree as the reviewers insightfuly pointed out that in the current work – designing an efficient learning system as if walking along an information space leverages knowledge on information geometry. Further, the authors would also like to highlight the coordinate independent nature of lagrangian formulation. After reading [1], the authors wonder whether such coordinate independence can be explained or connected with the invariance principle in IGO flow.
>
> **On related work at the intersection of classical mechanics and learning dynamics.** We thank the reviewers for pointing out the very relevant reference and would incorporate the discussion for each work in the next version.
>
> We thank the reviewer again for the helpful suggestions and believe that incorporating them will both clarify our conceptual positioning and situate the work more firmly within the broader ML context.
>
> > Re-deriving algorithms through the lens of Lagrangian dynamics as a latent unifying principle in itself is highly thought-provoking, while the re-derivation is one step, how do you envision to leverage the developed formalism going forward to aid in the development of new algorithms?
>
> We thank the reviewer for the very insightful question, and exactly re-derivation is the first step.
>
> We see the potential next steps for this work as three-fold:
>
> First, it opens the community with a starting point on the investigation of a principled way to derive learning rules from first principles rather than ad hoc design, tying together ideas from control theory, information geometry, and statistical inference, variational principles.
>
> Second, it offers a language for analyzing learning dynamics using tools familiar to physicists—action functionals, symmetries, conservation-like quantities—opening the door to connections with physics and AI.
>
> Third, through further investigating the Lagrangian dynamics, it could generate concrete, testable predictions about optimization behavior and sample efficiency (e.g., when and why certain optimizers outperform others with consideration of designing kinetic energy terms), which can guide both algorithm design and empirical study in modern deep learning.
>
> > Can the authors’ formalism, and posited principle of least action, be extended to more advanced forms of differentiation such as natural gradients, and the resulting algorithms such as natural gradient descent?
>
> We thank the reviewer for the very interesting pointer. We agree that natural gradient methods are highly relevant in light of our unifying perspective. Currently, the authors think the invariance principle may be an angle to investigate, if transformation Lagrangian is coordinate independent which likely involves use of information geometry and Natural Gradient. In the present work we focused on Adam and RMSprop as examples of adaptive Euclidean preconditioning. Extending the action to use an information metric opens a promising line of work toward a common variational theory that includes Adam, RMSprop, and natural gradient methods as different choices of metric.
>
> In conclusion, the authors regard the paper in its current form as conceptually intriguing for both theoretically driven and methodology focused ML audience. There are many natural next steps suggested by this framework, several of which we see as important directions for future work but are challenging to fully address within the scope of a single paper, given the complete novel nature of the approach. We would therefore very much welcome further discussion and feedback from the community.

---

### Meta-Review · Area_Chair_2uFK · 2025-12-19

**Summary:**

The paper proposes a unified theoretical framework that interprets learning as a physical process obeying the principle of least action, deriving algorithms like A-optimality, Bellman's equation, and Adam as stationary paths of a "Learning Lagrangian."

The main concern shared by the reviewers is that the paper’s central premise---ie interpreting efficient learning as a physical process that minimises a Lagrangian---lacks novelty, as this analogy has been previously established by researchers such as Naftali Tishby. Because this "connection" is not original to the authors, it cannot be considered as the primary contribution of the paper, regardless of whether the omission of prior work was intentional. With this central novelty removed, the remaining contributions were widely viewed as insufficient for acceptance at ICLR. In particular, there is an absence of experimental validation or novel theoretical results. Additionally, technical derivations, such as the claim that the framework derives the Adam optimiser, were criticised as distinct from the actual results (which resembled Natural Gradient Descent NGD instead).

In light of these criticalities, I cannot recommend this paper for acceptance.

**Reviewer Concerns:**

- **Reviewer hTkZ**:
- - Rating: 6 / Confidence: 3
- - The reviewer found the paper well-written and the unifying viewpoint "intriguing". However, they expressed concerns about the practical implications. In particular, the reviewer was concerned it lacks practical impact.
- - The authors response highlighted the possible development that the Langrangian Learning viewpoint can bring.
- - However, this contribution is highly contested and, although probably without knowing, was already established.
- **Reviewer 2i1D**:
- - Rating: 2 / Confidence: 5
- - The reviewer reported that the core analogy between learning algorithms and Lagrangian mechanics is not novel and was previously elaborated by Naftali Tishby. They also criticised the imprecise writing style.
- - The authors responded that they cited the specific Bellman derivation as textbook material and claimed their novelty lay in the unifying hypothesis and two new Lagrangians. They strongly objected to the reviewer's mention of "potential plagiarism"
- - This concern is still outstanding. The reviewer maintained that the core idea itself was being presented as a contribution rather than background.
- **Reviewer Sh1j**:
- - Rating: 8 / Confidence: 4
- - This reviewer was the most positive, appreciating writing and rigour. Their main critiques were the failure to connect with some existing work on information geometry and other physics-based ML approaches.
- - The authors promised to incorporate on the suggested references.
- **Reviewer N1Zv**:
- - Rating: 2 / Confidence: 2
- - The reviewer found the paper difficult to read (in particular Sec.2) and doubted about the derivation of Adam.
- - The authors clarified that Sec.2 was preparation material and admitted that the full Adam algorithm is not strictly derived from the simple Lagrangian.
- **Reviewer Szcw**:
- - Rating: 2 / Confidence: 3
- - The reviewer criticised the clarity and lack of standard notation. They also criticised the derivation of Adam, stating that instead that derivation leads to NGF.
- - The authors argued that their kinetic energy formulation distinguishes their approach from standard NGD.
- - The concerns about clarity and the derivation of Adam (vs. NGD) remain outstanding.

**Reviewer Scores:**

This part is completely speculative, as none of the reviewers explicitly announced a score update. Based on the discussion (summarised above), the reviewers may update their score as follows:

- **Reviewer hTkZ: 6 -> 5**: Given that the reviewer explicitly asked for practical uses to strengthen the paper and the authors confirmed none currently exist, the reviewer would likely lower their score slightly to reflect the confirmed lack of practical impact.

- **Reviewer 2i1D: 2 -> 2**: Given the severity of the dispute and the reviewer's attempt to alert other reviewers to the issue, their score would almost certainly remain a strong reject.

- **Reviewer Sh1j: 8 -> 8**: The only critiques of the reviewer were missing connections to Information Geometry and other physics-based works, and the authors promised to incorporate the references.

- **Reviewer N1Zv: 2 -> 3**: They might raise their score slightly for the engagement, but the core technical issues remain.

- **Reviewer Szcw: 2 -> 2**: Without a revised manuscript to prove that the clarity issues were fixed, the reviewer is unlikely to change their score.

---

### Decision · Program_Chairs · 2026-01-26

Reject